# Relationship Between Body Composition and Physical Literacy in Chilean Children (10 to 16 Years): An Assessment Using CAPL-2

**DOI:** 10.3390/jcm13237027

**Published:** 2024-11-21

**Authors:** Nicolás Muñoz-Urtubia, Alejandro Vega-Muñoz, Guido Salazar-Sepúlveda, Nicolás Contreras-Barraza, María Mendoza-Muñoz, Wladimir Ureta-Paredes, Remik Carabantes-Silva

**Affiliations:** 1Instituto de Ciencias de la Educación, Universidad Austral de Chile, Valdivia 5090000, Chile; nicolas.munoz01@uach.cl; 2International Graduate School, University of Extremadura, 10003 Cáceres, Spain; 3Facultad de Medicina y Ciencias de la Salud, Universidad Central de Chile, Santiago 8330507, Chile; alejandro.vega@ucentral.cl; 4Facultad de Ciencias Empresariales, Universidad Arturo Prat, Iquique 1110939, Chile; 5Facultad de Ingeniería, Universidad Católica de la Santísima Concepción, Concepción 4090541, Chile; gsalazar@ucsc.cl; 6Facultad de Ingeniería y Negocios, Universidad de Las Américas, Concepción 4090940, Chile; 7Pontificia Universidad Católica de Valparaíso, Valparaíso 2340025, Chile; 8Faculty of Sport Science, University of Extremadura, 10003 Cáceres, Spain; mamendozam@unex.es; 9Departamento Provincial de Educación Santiago Sur, Ministerio de Educación, Santiago 8910132, Chile; wladimir.ureta@mineduc.cl; 10Facultad de Ciencias Naturales, Matemática y del Medio Ambiente, Universidad Tecnológica Metropolitana, Santiago 8330383, Chile; rcarabantes.upa@utem.cl

**Keywords:** sport sciences, physical activity, motivation, knowledge, quality education, behavioral studies

## Abstract

**Objectives:** The aim of this study was to evaluate the levels of physical literacy (PL) in school children (10 to 16 years) in Santiago, Chile, and to explore the relationship between PL and body mass index (BMI). In addition, gender and age differences were analyzed in relation to PL levels in a context where childhood obesity remains a public health challenge. **Methods:** A total of 439 students in grades 5–8 were assessed using the Canadian Assessment of Physical Literacy-2 (CAPL-2) questionnaire. Demographic variables (sex, age, and BMI) were correlated with levels of motivation to engage in physical activity and physical activity knowledge. Chi-square and effect size were applied. **Results:** The results revealed a significant correlation between motivation for physical activity and gender, with boys showing higher levels of motivation than girls (χ² = 12.403, *p* < 0.006). In addition, an inverse relationship was observed between BMI and motivation (effect size = 0.198), suggesting that more motivated children tend to have a healthier BMI. Knowledge about physical activity increased with age (χ² = 60.460, *p* < 0.001) but did not have a significant influence on BMI. **Conclusions:** The findings highlight the need to design gender-specific interventions that enhance motivation as a key factor in promoting a healthy lifestyle and physical activity adherence. Public health strategies should include motivation-driven approaches to foster physical literacy and long-term engagement in physical activity, particularly for girls, to address pediatric health challenges in Chile. Public health policies should address these factors to improve pediatric health outcomes.

## 1. Introduction

There is a consensus in the scientific community on the definition of Physical Literacy (PL), which is described as the physical ability, motivation, confidence, knowledge, and willingness to participate in physically active lifestyles, encompassing four domains: physical, psychological, social, and cognitive [1,2,3,4,5]. It has been noted that the development of movement and sports skills encourages the adoption of more active and healthy behaviors, allowing a physically literate child to perform with ease and confidence in a variety of physically challenging situations, being able to react appropriately to a wide range of scenarios [1,3,6,7]. In contrast, those children who do not reach an adequate level of PL tend to avoid physical activity situations due to lack of confidence and motivation [8].

Internationally, PL has been recognized as crucial to physical education and public health, especially in Canada, Australia, and the United Kingdom, where research highlights its role in promoting active lifestyles and improving well-being [9,10]. Studies in these countries underline the importance of PL in engaging in physical activity throughout life [11]. In these countries, education and public health policies have integrated PL into school curricula, highlighting its importance in both school and out-of-school contexts [3,12]. Research demonstrates that PL fosters lifelong engagement in physical activity and contributes significantly to physical and psychological well-being, creating a foundation for active lifestyles that persist throughout all stages of life [13,14]. This international emphasis on PL underscores the importance of examining its role in developing contexts such as Chile, where childhood obesity and low levels of physical activity present significant health challenges [15,16].

The PL construct can be addressed in both educational and extracurricular settings. Castelli and Centeio [17] highlight that, within the educational context, curricula can support the development of PL in various ways, such as by distinguishing between structured, unstructured, or informal physical activities (such as recess) and through the teaching of physical activities enriched with academic content that integrates theoretical concepts and movement learning. Consequently, multiple investigations have explored the role of PL both in physical education classes [18,19,20] and in activities performed outside school hours [21,22,23]. The growing interest in PL and the benefits associated with its strengthening have prompted the development of assessment methods that facilitate its monitoring and control. An example of this is the Canadian Assessment of Physical Literacy (CAPL) [24], one of the first tools created for this purpose, which began to be developed in 2009 in response to the need to obtain objective data on PL. This tool was designed to be valid, reliable, feasible, and informative, with the purpose of assessing PL in Canadian children, covering several domains, such as fundamental motor skills, physical activity behavior, physical fitness, and knowledge, as well as awareness and understanding. The CAPL-2 provides a valid and reliable assessment of several PL domains, allowing for standardized comparison

Not only does PL generate and encourage greater participation in physical activity and greater adherence to a healthy lifestyle, but it also has a significant impact on the overall health of individuals [25]. An adequate level of PL is associated with a reduction in the prevalence of childhood obesity, because it promotes active behaviors and lifestyle, along with greater awareness and knowledge about the importance of performing and integrating the practice of regular physical activity into daily life [26,27]. The latter is particularly relevant, since childhood obesity is a significant risk factor for the development of chronic diseases such as type 2 diabetes [28].

PL contributes to the development of basic motor skills, which are fundamental during childhood, as these skills form the basis for participation in a wide range of physical activities throughout life [29]. A study by Huang et al. [30] indicated that children with a higher level of PL showed an improvement in basic motor skills, which in turn correlated with a higher rate of participation in physical activities and better overall physical fitness. From a health perspective, higher levels of PL are also related to improvements in cardiovascular and metabolic health. Recent research suggests that physically literate children are at lower risk of developing cardiovascular risk factors, such as hypertension and dyslipidemia, due to their higher level of regular physical activity [31,32].

Although PL has been extensively studied worldwide, evidence from Latin America remains limited, and the specific link between PL and BMI in this region has not been sufficiently explored [33]. This study aims to address these gaps by establishing the levels of LP among primary school children in Santiago, Chile, and analyzing its relationship with BMI, providing crucial data for public health strategies [34,35].

Finally, PL also has a positive impact on children’s mental health and psychological well-being, improving self-esteem and reducing symptoms of anxiety and depression, which contributes to an overall health status [36,37]. Due to the relevance of studying the development of PL, this study proposes as objectives (1) to establish the levels of PL in basic (primary) education children in Santiago, Chile, (2) to explore the relationship between PL and BMI in this Chilean population, and (3) to analyze the possible differences in PL levels according to the different BMI categories.

Consequently, correlational alternative hypotheses to theoretically and statistically evaluate aspects of this problem are proposed:**H1.1:** *Sex is not independent, or there is an association with the motivation level (MOTLEV).***H1.2:** *Sex is not independent, or there is an association with the knowledge level (KNOLEV).***H2.1:** *The nutritional category (NUTCAT) is not independent, or there is an association with the motivation level (MOTLEV).***H2.2:** *The nutritional category (NUTCAT) is not independent, or there is an association with the knowledge level (KNOLEV).***H3.1:** *Age is not independent, or there is an association with motivation level (MOTLEV).***H3.2:** *Age is not independent, or there is an association with the knowledge level (KNOLEV).***H4.1:** *Age is not independent, or there is an association with the nutritional category (NUTCAT).*

## 2. Materials and Methods

To achieve our research objectives, we used the CAPL-2 questionary in Spanish (applying a semantic validation procedure by 10 Chilean primary students, conducted by 3 co-authors who validated the scale in Spain) [38] and the BMI parameters (height and body weight; data detailed in Appendix A). BMI was calculated from direct height and weight measurements. For the estimation of nutritional status, based on BMI, the ‘growth standards’ of the Chilean Ministry of Health [39] were used, considering absolute ages from 10 to 16 years (in integer years). The surveyed population corresponds to 439 students from 5th to 8th grade of primary education from 4 socioeconomically equivalent (Middle Class) educational establishments in the Santiago Sur sector (Chile), 2 establishment cases in the El Bosque municipality, and 2 establishment cases in the La Cisterna municipality. All students present in class on the day of sampling were included, provided they had informed consent from their parents and personal informed assent, as indicated in the Institutional Review Board Statement declaration. The participating students’ characteristics are presented in Table 1, by variables: sex (ordinal type), nutritional category (ordinal type), and age (scale type).

The results have been obtained from the CAPL-2 questionnaire [38], complementing the results in older students according to the study of Blanchard et al. [40], in relation to the level of motivation (MOTLEV) for physical activity and the level of knowledge (KNOLEV) about physical activity. Using SPSS software version 23 (IBM, New York, NY, USA), responses have been contrasted with the demographic variables (sex, nutritional category, and age) by means of a nonparametric statistical analysis of chi-squared test, determining the degree of dependence between two variables using a *p*-value, whose acceptable significance is <0.050 and good significance is <0.010 [41,42].

We used the chi-square test in this research to assess the relationship between categorical variables, specifically demographic variables (such as sex, nutritional category, and age) and levels of motivation and knowledge about physical activity in Chilean children. This test is particularly suitable for our analysis because it allows us to identify whether there is a significant association between these categories, helping us quantify the magnitude of these associations and providing a solid statistical basis for interpreting the results and proposing specific interventions [43]. For results with significance in the chi-squared test, the effect size sensitivity using G*Power version 3.1.9.7 is calculated (Heinrich Heine University Düsseldorf, Düsseldorf, Northrhine-Westphalia, Germany), considering the following cut-off points for effect sizes: small = 0.1; medium = 0.3; and large = 0.5 [44] (see Table 2).

## 3. Results

The results of the demographic variables regarding sex, nutritional category (NUTCAT), and age, were correlated with the motivation level (MOTLEV) for physical activity and the knowledge level (KNOLEV) about physical activity (see Table 3).

Therefore, in this case, motivation level correlates well with sex and acceptably with nutritional category, and knowledge level correlates well with age, although all of these have a small effect size. Additionally, nutritional category correlates acceptably with age (*p*-value = 0.032; degree freedom = 12; and effect size = 0.245). All significant correlations can be seen in Figure 1.

Figure 1a shows the sexual differences, to the detriment of the female gender, in motivation to participate in physical activities, where boys present higher levels of motivation to engage in physical activity compared to girls. Figure 1b reveals that as nutritional category increases, motivation to engage in physical activity decreases, especially in students with higher BMI. Similarly, Figure 1c illustrates how the level of knowledge about physical activity increases with age, reflecting greater knowledge in students with greater age, i.e., upper grades. In relation to the nutritional category, Figure 1d indicates that being overweight is more prevalent as age increases, with an increase in the proportion of overweight children in the older age groups.

## 4. Discussion

The results of our study show different interesting dynamics in the relationship between physical literacy, BMI, and sociodemographic factors such as age and sex in children and adolescents who participated in the study. These findings are explored in detail below, organized into key points.

### 4.1. Relationship Between Age and BMI

We observe that as children grow older, their BMI tends to increase. The global trend for childhood obesity continues to rise [45]. This finding could be related to changes in lifestyle as age increases, which could be determined by the increase in high-calorie foods during adolescence [46] and the decrease in physical activity during this same age, because children and adolescents decrease their rate of physical activity as they increase in age due to commitments related to increased academic difficulty along with increased screen time [47].

In our study, this difference is appreciated as we observe the prevalence of an increase in BMI in older age groups, which is in line with what has been observed at the national level in Chile where the trend is towards increasing obesity. For example, in the 2016–2017 National Health Survey, almost 44% of children between 5 and 17 years old were identified as overweight or obese [48]. This tendency is further emphasized by Chile’s 2022 report card on physical activity for children and adolescents [49], which highlights that Chilean youth tend to reduce their physical activity levels as they grow older, supporting the increasing trend of BMI as age increases in the pediatric population.

However, it is important to highlight that, in terms of nutritional categories by BMI, the youngest children in our study presented a higher prevalence of overweight and obesity, despite evidence showing that BMI generally increases with age. This contrasts with national data [50], which show that 5th graders present the highest prevalence of overweight and obesity.

These results highlight the need for age-specific interventions to address increasing BMI and promote active lifestyles in children. Programs such as those implemented in the United Kingdom [51] and Australia [52] have successfully used age-specific strategies to mitigate the increase in BMI by promoting physical activity and healthier eating habits. Similar approaches in Chile, adapted to different stages of development, could increase the impact and effectiveness of public health policies aimed at reversing childhood obesity trends.

It is also essential to note that BMI, while widely used, may not capture the full range of body composition changes, such as muscle mass or fat distribution, in children. Future research should consider additional indicators for a more comprehensive understanding [53].

### 4.2. Motivation and Sex

Motivation to participate in physical activity, on the other hand, is inversely related with sex, with most boys increasing or maintaining their motivation to engage in physical activity as they get older, unlike their female counterparts, among whom this trend was not evident. In this sense, and in view of the results, the need for specific interventions to address gender differences in motivation for physical activity, especially in girls and adolescents, is suggested [54]. In this regard, Jekauc et al. [55] found that gender differences in motivation for physical activity widen with age, with boys maintaining higher levels of motivation than girls throughout adolescence. At the same time, the 2018 National Survey of Physical Activity and Sports in Chile supports this observation, concluding that men tend to participate more in physical activities than women, especially during adolescence, which could suggest the presence of cultural and social barriers that could undermine women’s motivation to engage in physical activity [56]. This pattern is also observed in Brazil, where studies such as Alemany’s [57] show higher levels of motivation and participation in sports activities by boys than by girls. This disparity is attributed to social expectations and gender stereotypes that link physical activity with masculinity, which limits female participation in these areas.

Similarly, research conducted in Mexico [58] reveals that girls are less likely to engage in physical activity due to the limited encouragement they receive from the family and school environments, both of which are shaped by gender norms that favor boys’ participation in sports.

Likewise, in the United States, girls encounter additional barriers to physical activity, such as body image concerns and lack of role models, which may decrease their motivation to be active [59]. These findings highlight the need for targeted interventions that create supportive environments for girls, challenging gender stereotypes and fostering sustained motivation for physical activity in a variety of contexts.

### 4.3. Knowledge and Age

Levels of knowledge about physical activity increase with age and are independent of sex. This increase may be related to prolonged exposure to the educational environment, where the benefits of physical activity are consistently promoted [60]. Furthermore, as adolescents develop self-reflection and critical thinking skills, their ability to understand and analyze health concepts from a holistic perspective also increases, fostering a more complex and comprehensive understanding of the benefits of physical activity [61]. This cognitive development allows them to critically engage with health messages, transforming information into meaningful insights. In this context, educational factors such as curriculum structure and targeted health education programs play a fundamental role in reinforcing physical literacy concepts and contributing to cumulative knowledge over time [62,63]. This finding aligns with the conceptualization of physical literacy as a progressive journey across the lifespan, where knowledge accumulates over time and experience [64].

Both The National Health Survey 2016–2017 [48] and “Chile’s 2022 report card on physical activity for children and adolescents” [49] have reported an increasing awareness of the importance of physical activity in Chile. However, this knowledge does not always translate into healthy practices, supporting our observation that knowledge alone does not directly influence the nutritional category. One explanation may be that, although knowledge provides a basis, turning awareness into action requires intrinsic motivation together with a supportive social environment. In this sense, Bandura [65] explains that social support and positive reinforcement play a determining role in the adoption and maintenance of behaviors, including health-oriented behaviors. In this sense, Bailey et al. [66] highlight that, although knowledge is an essential component of physical literacy, its impact on healthy behavior is mediated by motivation and the social environment.

Therefore, interventions aimed at improving physical literacy should not only focus on knowledge acquisition but should also integrate strategies to foster intrinsic motivation and a supportive social context [67]. For example, incorporating goal-setting and self-efficacy exercises could actively engage adolescents, while linking family and community support could reinforce healthy behaviors in younger children [68]. These approaches could help bridge the gap between knowledge and behavior, ensuring that the benefits of physical literacy translate into lasting, lifelong healthy habits [69].

### 4.4. Relationship Between Motivation and Nutritional Category

The inverse relationship found between motivation for physical activity and BMI suggests that more motivated children tend to have a BMI closer to that expected for their age and height. This finding underscores the importance of motivation as a key factor in promoting a healthy weight and lifestyle, highlighting the need to foster a culture that motivates children and adolescents to be active [70]. Owen et al. [71] found that intrinsic motivation to engage in physical activity is associated with greater adherence to a healthy lifestyle and, therefore, a lower incidence of obesity and overweight in adolescents.

### 4.5. Knowledge and Nutritional Status

Interestingly, we did not find a significant relationship between knowledge level and nutritional category. This suggests that, although knowledge about the benefits of regular physical activity is important, it is not sufficient by itself to influence behaviors that affect BMI [72].

This finding could be explained by the fact that knowledge is not always translated into action. Psychosocial factors such as family context, the availability of healthy foods, and the social and cultural environment may play a determinant role in nutritional decisions [65,66,73]. Similarly, the lack of practical skills to apply this knowledge in daily life could limit its impact on nutritional status, creating a gap between theory and daily choices [74].

Likewise, it supports the idea that, although knowledge about physical activity is included within the concept of physical literacy, its influence on health will also depend on motivational and behavioral factors that need to be addressed in a comprehensive manner [75]. Along these lines, the suggestion is that physical activity and healthy lifestyle education should be accompanied by motivational strategies that foster a supportive environment to ensure the effectiveness of interventions [69,76].

Likewise, improvement in physical literacy levels also requires improvements in nutritional labeling [77] so that children and adolescents can make informed decisions about their own nutrition. Muñoz et al. [78] in their study on the influence of body composition on physical literacy in Spanish children highlights that a higher percentage of fat mass is associated with lower levels of physical competence, which in turn could be linked to increased consumption of junk food and energy drinks as children get older [46,77].

### 4.6. Implications for Public Health

The results of our study, together with data from national surveys, underscore the importance of public health policies that address both education and motivation for physical activity, that is, physical literacy as a whole [79]. For their part, the results of the 2016–2017 National Health Survey and Chile’s 2022 report card on physical activity for children and adolescents highlight the urgent need to intervene early in life to reverse the increasing trends of overweight and obesity, which reinforces the relevance of our findings in the Chilean context [48,49]. Specific recommendations based on our findings include the development of school-based physical education programs that promote practical skills and intrinsic motivation, both tailored to the needs of different age groups. Age-specific interventions have been shown to be particularly effective in improving physical literacy and supporting long-term healthy behaviors in children and adolescents [80]. Primary programs could focus on basic movement skills, while secondary programs could emphasize personal goal setting and self-management [81,82]. Interventions should be specific for each age group and sex, considering the barriers and motivations of each subgroup to maximize their effectiveness [83,84]. Due to the observed motivational differences between males and females, policies should also aim to reduce gender disparities in physical activity participation by creating supportive and inclusive environments specifically for girls [85].

### 4.7. Considerations for Future Research

Future research should consider confounding variables like socioeconomic status, access to nutrition, and parental support, as these factors can influence BMI and add context to the relationship between age and BMI [86,87]. Including additional body composition indicators, such as waist-to-height ratio [88] or bioelectrical impedance analysis [89], can provide a more nuanced understanding of body composition changes that BMI alone cannot capture, particularly in differentiating fat distribution and lean mass. Longitudinal designs exploring motivation and social support across different ages and genders would further refine understanding and enhance public health interventions for young populations [90].

## 5. Conclusions

Our study provides information on the relationships between physical literacy, BMI, age, and gender in Chilean youth. While BMI tends to increase with age, the reasons behind this trend, potentially including lifestyle changes, are complex and require further exploration. Knowledge about physical activity also increases with age but does not seem sufficient on its own to positively influence BMI, and it is likely that motivation, which is generally higher in males, plays a more crucial role. These results suggest the value of age- and sex-specific programs to effectively promote physical literacy in the child and youth population.

The results demonstrate practical implications for educators, health professionals, and policy makers to support active and informed lifestyles in young populations. Future research could examine broader sociodemographic influences, such as socioeconomic status, and include additional measures of body composition to improve intervention strategies.

## Figures and Tables

**Figure 1 jcm-13-07027-f001:**
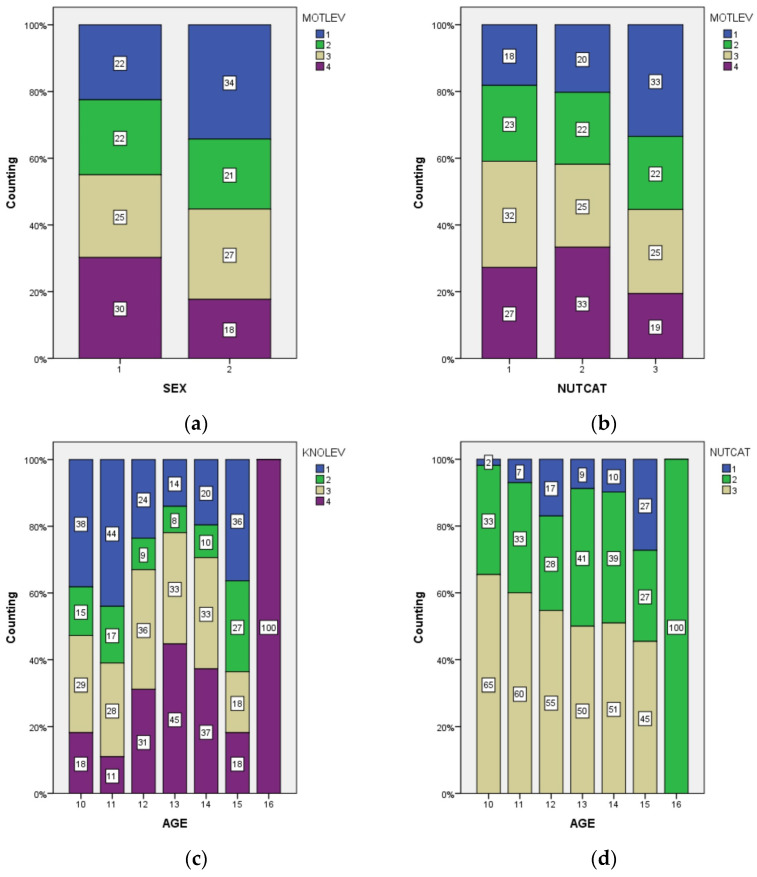
Relation between demographic variables and PL variables: (**a**) sex and motivation level, (**b**) nutritional category and motivation level, (**c**) age and knowledge level, and (**d**) age and nutritional category.

**Table 1 jcm-13-07027-t001:** Characterization of sample.

Demographic Variable	Category		N		%	
Sex	Male (1) *		258		58.8%	
	Female (2)		181		41.2%	
Nutritional Category (NUTCAT)	Underweight (1)		44		10.0%	
	Normal weight (2)		153		34.9%	
	Overweight (3)		242		55.1%	
Age (in years)	Child (1)	10	261	55	59.4%	12.5%
		11		100		22.8%
		12		106		24.1%
	Adolescent (2)	13	178	114	40.6%	26.0%
		14		51		11.6%
		15		11		2.5%
		16		2		0.5%

* The numbers in parentheses correspond to the coding assigned in the database.

**Table 2 jcm-13-07027-t002:** Cut-off for *p*-value and effect sizes.

Test	Cut-Off 1	Cut-Off 2	Cut-Off 3
*p*-value	<0.050 (acceptable)	<0.010 (good)	
Effect size	0.10 (small)	0.30 (medium)	0.50 (large)

**Table 3 jcm-13-07027-t003:** Chi-square tests: demographic variables and PL.

Demographic Variables	Parameters	MOTLEV	KNOLEV
Sex	Value	12.403	6.650
	Asymptotic significance (bilateral)	0.006 **	0.084
	Degree freedom	3	3
	Effect size (power 95%)	0.198 *	0.198 *
NUTCAT	Value	15.547	11.768
	Asymptotic significance (bilateral)	0.016 *	0.070
	Degree freedom	6	6
	Effect size (power 95%)	0.212 *	0.212 *
Age	Value	24.114	60.460
	Asymptotic significance (bilateral)	0.151	0.000 **
	Degree freedom	18	18
	Effect size (power 95%)	0.230 *	0.230 *

Asymptotic significance (bilateral): ** good fit and * acceptable fit. Effect size (power 95%): * small.

## Data Availability

The data are original and were delivered with the article as Appendix A (PLBM_v02.csv).

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
