# Peer review of "Relationship Between Body Composition and Physical Literacy in Chilean Children (10 to 16 Years): An Assessment Using CAPL-2"

_jcm, 2024, doi:10.3390/jcm13237027_

Round 1
Reviewer 1 Report
Comments and Suggestions for Authors
Dear Editor,
Thank you for the opportunity to review the manuscript "Relationship Between Body Composition and Physical Literacy in Chilean Children: An Assessment Using CAPL-2" for the Journal of Clinical Medicine. This study explores the levels of physical literacy among primary school children in Santiago, Chile, and investigates the relationships between physical literacy, body mass index (BMI), age, and gender. The main findings suggest that motivation for physical activity is inversely related to BMI and differs by gender, with boys showing higher motivation levels. Knowledge about physical activity increases with age but does not significantly influence BMI. Overall, this manuscript provides valuable insights into the complex relationships between physical literacy components and body composition in Chilean children, with potential implications for public health interventions.
General Comments
1. Theoretical Framework: The introduction could benefit from a more comprehensive theoretical framework linking physical literacy to health outcomes, particularly in the context of developing countries.
2. Methodology: The authors should provide more details about the CAPL-2 questionnaire, including its validity and reliability in the Chilean context.
3. Results: The presentation of results could be enhanced with more detailed statistical analyses, including effect sizes and confidence intervals where appropriate.
4. Discussion: While the discussion covers several important points, it could be strengthened by more in-depth analysis of the findings in relation to existing literature and the unique context of Chile.
5. Implications: The authors should elaborate on the practical implications of their findings for public health policy and interventions in Chile and similar contexts.
6. Limitations: A more comprehensive discussion of the study's limitations would strengthen the manuscript.
Specific Comments
Title
· The title accurately reflects the content of the study. However, consider specifying the age range of the children studied to provide more immediate context.
Abstract
· Clarify the specific age range of the "primary school children" studied.
· Include brief details on the sample size and demographics.
· Provide more specific results, including key statistical findings.
· Expand on the implications and conclusions to better highlight the study's significance.
Introduction
· Provide a more comprehensive definition of physical literacy, citing key literature in the field.
· Expand on the relevance of studying physical literacy in the Chilean context, particularly in relation to childhood obesity trends.
· Clarify the rationale for using the CAPL-2 assessment tool in this study.
· State the research hypotheses more explicitly.
Materials and Methods:
· Provide more details about the selection process for the 4 educational establishments and any inclusion/exclusion criteria for participants.
· Explain how the Spanish translation of CAPL-2 was validated for use in Chile.
· Describe the BMI measurement process and the criteria used for categorizing nutritional status.
· Justify the choice of chi-square tests for analysis and explain how the p-value thresholds were determined.
· Include information on ethical approval and informed consent procedures.
Results
· Present more detailed statistical results, including effect sizes and confidence intervals where appropriate.
· Consider including a correlation matrix to show relationships between all variables studied.
· Improve the quality of Figure 1 by using color to differentiate categories and providing more detailed captions.
· Consider adding tables to present the descriptive statistics and chi-square test results in full.
Discussion
· Firstly, consider adding subheadings in the Discussion section to improve organization and readability.
· Expand the discussion on the relationship between age and BMI, considering potential confounding factors and how they were addressed.
· Provide more context for the gender differences in motivation. Compare these findings with other studies in Latin America or globally.
· Analyze the increase in knowledge with age more deeply, discussing potential mechanisms and implications for physical literacy interventions.
· Explore potential explanations for the lack of relationship between knowledge and nutritional status, considering implications for health education strategies.
· Discuss the limitations of using BMI as the sole measure of body composition in children.
Implications for Public Health
· Provide more specific recommendations based on the findings, particularly tailored to the Chilean context.
· Discuss how these findings could inform the design of physical education programs and health interventions in schools.
· Consider the broader societal and policy implications of the gender differences observed in motivation for physical activity.
Conclusions
· More clearly state the novel contributions of this study to the field of physical literacy and child health.
· Suggest specific directions for future research based on the findings and limitations of this study.
· Emphasize the practical implications of the findings for educators, health professionals, and policymakers.
In conclusion, this manuscript addresses an important topic and provides valuable insights into physical literacy and body composition in Chilean children. With the suggested revisions, particularly in strengthening the theoretical framework, expanding the discussion, and clarifying the implications, this paper has the potential to make a significant contribution to the field.
Author Response
Thank you very much for your observations, they have been very significant to improve our research. We will respond to each issue:
Title
1.- The title accurately reflects the content of the study. However, consider specifying the age range of the children studied to provide more immediate context.
R1. We have modified the title, considering the age range of the Chilean students (10 to 16 years)
Abstract
2.- Clarify the specific age range of the “primary school children” studied.
R2. In the abstract, the age of the participants has been specified (line 24).
3.- Include brief details on the sample size and demographics.
R3. In the abstract, the sample size and the tools used for the analysis, such as Chi-square and Effect Size, were included (line 29).
4.- Provide more specific results, including key statistical findings.
R4. The statistical results of the Chi-square test (lines 31 and 34) and the Effect Size (line 32) have been added.
5.- Expand on the implications and conclusions to better highlight the study's significance.
R5. It was pointed out that public health strategies should include motivation-based approaches (lines 36-37).
Introduction
6.- Provide a more comprehensive definition of physical literacy, citing key literature in the field.
R6. The Introduction was modified, adding literature and definitions to provide a more comprehensive explanation of physical literacy (lines 53-64).
7.- Expand on the relevance of studying physical literacy in the Chilean context, particularly in relation to childood obesity trends.
R7. The Introduction was updated with 3 articles indexed in WOS on physical literacy in Chile and Latin America (lines 99-101).
8.- Clarify the rationale for using the CAPL-2 assessment tool in this study.
R8. Specify the characteristics of the CAPL-2 questionnaire, which assesses multiple domains of physical literacy and allows comparisons between subjects (lines 80-81).
9.- State the research hypotheses more explicitly.
R9. Seven explicit hypotheses have been added (lines 111-126).
Materials and Methods:
10.- Provide more details about the selection process for the 4 educational establishments and any inclusion/exclusion criteria for participants.
R10. Details were added about the characteristics of the selected establishments, which are public and have a similar socioeconomic profile (lines 134-138).
11.- Explain how the Spanish translation of CAPL-2 was validated for use in Chile.
R11. The validation of the Spanish translation of CAPL-2 was performed through a semantic validation process (lines 128-130).
12.- Describe the BMI measurement process and the criteria used for categorizing nutritional status.
R12. BMI was measured directly, and the growth charts of the Chilean Ministry of Health were used to categorize nutritional status (lines 131-133).
13.- Justify the choice of chi-square tests for analysis and explain how the p-value thresholds were determined.
R13. The use of the chi-square test was justified, since it evaluates the association between categorical variables by comparing the observed distribution with the expected one, assuming independence between the variables. The p-value thresholds were set at the common significance level of 0.05 (lines 154-160).
14.- Include information on ethical approval and informed consent procedures.
R14. Information on ethical approval and informed consent procedures was added to the methodology (lines 138-140).
Results
15.- Present more detailed statistical results, including effect sizes and confidence intervals when appropriate.
R15. Statistical results are presented in detail in Table 3.
16.- Consider including a correlation matrix to show the relationships between all the variables studied.
R16. The relationships between all the variables studied are detailed in Table No. 3.
17.- Improve the quality of Figure 1 by using colors to differentiate the categories and providing more detailed legends.
R17. The colors in Figure 1 help to differentiate the categories of results.
18.- Consider adding tables to present descriptive statistics and chi-square test results in their entirety.
R18. The descriptive and chi-square test results are presented in detail in Table 3.
Discussion
19.- Firstly, consider adding subheadings in the Discussion section to improve organization and readability.
R19. The Discussion section has seven subheadings (lines 195, 227, 253, 285, 315, 334).
20.- Expand the discussion on the relationship between age and BMI, considering potential confounding factors and how they were addressed.
R20. It is specified that the younger the age, the overweight nutritional category is observed, while the older the age, the higher the proportion of normal weight. In addition, it is mentioned that BMI does not measure changes in the composition of fat and muscle mass (lines 211-226).
21.-Provide more context for the gender differences in motivation. Compare these findings with other studies in Latin America or globally.
R21. Information from studies in Brazil, Mexico and the United States was included to contextualize the gender differences in motivation (lines 239-252).
22.-Analyze the increase in knowledge with age more deeply, discussing potential mechanisms and implications for physical literacy interventions.
R22. Information was added on curricula and health programs that keep children informed over time (lines 256-264). In addition, the critical role of social support and positive reinforcement in establishing healthy habits was highlighted (lines 271-275) and it was suggested that interventions focus on the development of health habits and goals (lines 278-284).
23.- Explore potential explanations for the lack of relationship between knowledge and nutritional status, considering implications for health education strategies.
R23. The term “nutritional status” was replaced by “nutritional category”. The lack of relationship may be due to family, cultural and dietary factors (lines 298-302).
24.-Discuss the limitations of using BMI as the sole measure of body composition in children.
R24. In the recommendations for future research, it was considered that BMI alone does not measure changes in body composition (lines 211-226).
Implications for Public Health
25.- Provide more specific recommendations based on the findings, particularly tailored to the Chilean context.
R25. Specific recommendations for physical literacy programs were added (lines 326-333).
26.- Discuss how these findings could inform the design of physical education programs and health interventions in schools.
R26. It was considered necessary to implement age-specific interventions in physical education programs (lines 322-328).
27.- Consider the broader societal and policy implications of the gender differences observed in motivation for physical activity.
R27. It was recommended that specific programs be included to ensure the participation of women in physical activity (lines 330-333).
Conclusions
28.- More clearly state the novel contributions of this study to the field of physical literacy and child health.
R28. The novel contribution of this study in relation to physical literacy, BMI, age and gender in Chilean children was included (lines 349-350).
29.- Suggest specific directions for future research based on the findings and limitations of this study.
R29. It was suggested that future studies examine socioeconomic variables and include measures of body composition (lines 358-360).
30.- Emphasize the practical implications of the findings for educators, health professionals, and policymakers.
R30. It was emphasized that physical literacy initiatives should primarily consider sex and age range (lines 354-356).

Reviewer 2 Report
Comments and Suggestions for Authors
The manuscript addresses the relationship between physical literacy (PL), body mass index (BMI), and sociodemographic factors in a Chilean paediatric population. While the study is relevant and timely, several areas require critical improvement for clarity, scientific rigor, and structure.
Introduction
The introduction could benefit from a broader international context of PL, showing how it is defined and studied in different populations before honing in the Chilean context. This contextualisation would help justify why PL and its relationship with BMI are particularly relevant in Chile.
The transition to study objectives (PL levels, PL-BMI relationships, and differences by BMI category) lacks clear justification in the introduction. Explicitly linking gaps in the literature with the study’s objectives could strengthen the rationale.
Methodology
The methodology lacks details on the sampling strategy (e.g. how schools were chosen and sample representativeness of Santiago's broader population). There is also limited information on how 5th- to 8th-grade participants were selected. Specifying the inclusion/exclusion criteria and sampling methods would improve rigor.
The authors stated that they used the CAPL-2 questionnaire but lacked details on its adaptation to Spanish, its validity in Chilean children, or any adjustments made for cultural differences. Without this, there may be issues with the construct validity of PL in this population.
Discussion
Some sections of the discussion appear to repeat the findings from prior studies rather than critically interpreting the study’s results. For example, citing global trends in childhood obesity without a direct comparison to the study findings weakens the discussion.
The discussion does not sufficiently address why PL motivation decreases with age, particularly for girls, or how this could reflect cultural or societal factors specific to Chile. Furthermore, the relationship between age and knowledge can be discussed in light of developmental psychology or educational factors, providing a stronger theoretical basis.
The authors did not discuss potential confounding variables (e.g. socioeconomic status and physical environment) that could influence the relationship between PL and BMI. Adding these considerations would add depth to the discussion and acknowledge the limitations of this study.
The manuscript could benefit from a clearer articulation of how these findings might inform policy in Chile and similar Latin American contexts. For example, should school programs integrate PL assessments, or are policy changes needed at the national level?
- .
Conclusions
Rather than summarising all findings, the conclusion could be strengthened by focusing on the most significant results, such as the observed differences in PL by age and gender, and the implications for intervention.
Recommendation
Considering the issues outlined above, this manuscript should be reconsidered after major revisions. This study is relevant and has potential, but critical flaws in methodology, data interpretation, and presentation limit its current quality.
Comments on the Quality of English Languageenglish is fine
Author Response
Thank you very much for your observations, they have been very significant to improve our research. We will respond to each issue:
Introducción
1.- The introduction could benefit from a broader international context of PL, showing how it is defined and studied in different populations before honing in the Chilean context. This contextualisation would help justify why PL and its relationship with BMI are particularly relevant in Chile.
R1: We have added studies from other populations in the world such as Canada, Australia and the United Kingdom (Line 55 to Line 66).
2.- The transition to study objectives (PL levels, PL-BMI relationships, and differences by BMI category) lacks clear justification in the introduction. Explicitly linking gaps in the literature with the study’s objectives could strengthen the rationale.
R2: To improve the justification we have added studies by the authors Pavez-Adasme et al, Contreras - Zapata et al and YDO on the gap of this type of studies in Latin America specifically between PL and BMI (Line 99 to line 103).
Methodology
3-The methodology lacks details on the sampling strategy (e.g. how schools were chosen and sample representativeness of Santiago's broader population). There is also limited information on how 5th- to 8th-grade participants were selected.
R3: We have included in a better way the criteria of sample considerations, socioeconomic criteria were informed, we have also clarified that 2 educational establishments of the commune of El Bosque and 2 educational establishments of the commune of La Cisterna have participated, in addition we have generated an inclusion of students who attended classes on the day of sampling and had informed consent and assent (Line 135 to line 140).
4.- The authors stated that they used the CAPL-2 questionnaire but lacked details on its adaptation to Spanish, its validity in Chilean children, or any adjustments made for cultural differences. Without this, there may be issues with the construct validity of PL in this population.
R4: The questionnaire used is adapted for Spanish (in Spain) by our same team of researchers under the same procedure was re-adapted for Chilean Spanish (line 128 to line 130).
Discusión
5.- Some sections of the discussion appear to repeat the findings from prior studies rather than critically interpreting the study’s results. For example, citing global trends in childhood obesity without a direct comparison to the study findings weakens the discussion.
R5: We have complemented our discussion by critically emphasizing the different results of our study with others, specifically with the national health survey (line 211 to line 215).
6.- The discussion does not sufficiently address why PL motivation decreases with age, particularly for girls, or how this could reflect cultural or societal factors specific to Chile. Furthermore, the relationship between age and knowledge can be discussed in light of developmental psychology or educational factors, providing a stronger theoretical basis.
R6: We have complemented our discussion with a study of the effects of socio-environmental factors on motivation (Line 271 to Line 275).
7.- The authors did not discuss potential confounding variables (e.g. socioeconomic status and physical environment) that could influence the relationship between PL and BMI. Adding these considerations would add depth to the discussion and acknowledge the limitations of this study.
R7: The four schools included in the study correspond to the same socioeconomic level (middle class), as we have also reported in the methodological section (Line 136).
8.- The manuscript could benefit from a clearer articulation of how these findings might inform policy in Chile and similar Latin American contexts. For example, should school programs integrate PL assessments, or are policy changes needed at the national level?
R8: We have related our research findings to studies from Mexico, Brazil and the USA, which consider gender and motivation variables and could be considered in Chile's public policy (Line 239 to line 252).
Conclusions
9.- Rather than summarising all findings, the conclusion could be strengthened by focusing on the most significant results, such as the observed differences in PL by age and gender, and the implications for intervention.
R9: We have modified the way we report the findings to make clearer the most significant contributions, reinforced the differences between BMI and age, the reasons for the differences, knowledge and age, and suggestions for physical literacy intervention programs by gender and age and reinforced the practical implications (Line 345 to Line 347).

Round 2
Reviewer 1 Report
Comments and Suggestions for Authors
Dear Editor,
I have reviewed the authors' responses to my comments regarding their manuscript "Relationship Between Body Composition and Physical Literacy in Chilean Children: An Assessment Using CAPL-2". The authors have demonstrated a commendable commitment to improving their manuscript by addressing all points raised in the initial review.
The methodological enhancements are particularly noteworthy, including the detailed explanation of the CAPL-2 validation process for Chilean context and the comprehensive description of BMI measurement procedures. The addition of seven explicit research hypotheses has significantly strengthened the study's theoretical framework and analytical approach.
The statistical analysis has been substantially improved with the inclusion of detailed chi-square test results, effect sizes, and comprehensive data presentation in tables. The expanded discussion section, now organized with clear subheadings, offers a more nuanced interpretation of the findings, particularly regarding age-related differences in physical literacy and gender variations in motivation.
The authors have successfully contextualized their findings within the broader Latin American literature, particularly in relation to gender differences in physical activity motivation. The practical implications have been thoughtfully developed, with specific recommendations for physical education programs and public health initiatives in Chile.
The limitations section now provides a more balanced view of the study's constraints, particularly regarding the use of BMI as a measure of body composition. The future research directions suggested are both practical and relevant to advancing understanding in this field.
Based on the comprehensive nature of the revisions and the significant improvements made to the manuscript. The study makes a valuable contribution to understanding the relationship between physical literacy and body composition in Chilean children, with clear implications for public health and education policy.
Sincerely,
The reviewer
Reviewer 2 Report
Comments and Suggestions for Authors
the authors have addressed all my concerns adequately